# TRPV4 is the temperature-sensitive ion channel of human sperm

Nadine Mundt[1,2], Marc Spehr[2], Polina V Lishko[1]*

[1]Department of Molecular and Cell Biology, University of California, Berkeley, Berkeley, United States; [2]Department of Chemosensation, Institute for Biology II, RWTH Aachen University, Aachen, Germany

**Abstract** Ion channels control the ability of human sperm to fertilize the egg by triggering hyperactivated motility, which is regulated by membrane potential, intracellular pH, and cytosolic calcium. Previous studies unraveled three essential ion channels that regulate these parameters: (1) the $Ca^{2+}$ channel CatSper, (2) the $K^+$ channel KSper, and (3) the $H^+$ channel Hv1. However, the molecular identity of the sperm $Na^+$ conductance that mediates initial membrane depolarization and, thus, triggers downstream signaling events is yet to be defined. Here, we functionally characterize DSper, the Depolarizing Channel of Sperm, as the temperature-activated channel TRPV4. It is functionally expressed at both mRNA and protein levels, while other temperature-sensitive TRPV channels are not functional in human sperm. DSper currents are activated by warm temperatures and mediate cation conductance, that shares a pharmacological profile reminiscent of TRPV4. Together, these results suggest that TRPV4 activation triggers initial membrane depolarization, facilitating both CatSper and Hv1 gating and, consequently, sperm hyperactivation.
DOI: https://doi.org/10.7554/eLife.35853.001

## Introduction

The ability of human spermatozoa to navigate the female reproductive tract and eventually locate and fertilize the egg is essential for reproduction (*Okabe, 2013*). To accomplish these goals, a spermatozoon must sense the environment and adapt its motility, which is controlled in part by ATP production and flagellar ion homeostasis (*Lishko et al., 2012*). Of vital importance is the transition from symmetrical basal tail bending into 'hyperactivated motility' – an asymmetrical, high-amplitude, whip-like beating pattern of the flagellum - that enables sperm to overcome the egg's protective vestments (*Ho et al., 2002*; *Ishijima et al., 2006*; *Quill et al., 2003*; *Suarez, 1996*, *2008*, *1993*). The steroid hormone progesterone (P4) acts as a major trigger of human sperm hyperactivation (*Suarez, 2008*; *Uhler et al., 1992*). P4 is released by cumulus cells surrounding the egg (*Schuetz and Dubin, 1981*) and causes a robust elevation of human sperm cytoplasmic $[Ca^{2+}]$ via the principal $Ca^{2+}$ channel of sperm, CatSper ($EC_{50}$ = 7.7 ± 1.8 nM) (*Lishko et al., 2011*; *Strünker et al., 2011*; *Smith et al., 2013*). The steroid acts via its non-genomic receptor ABHD2, a serine hydrolase that, upon P4 binding, releases inhibition of human CatSper (*Miller et al., 2016*). The $Ca^{2+}$ influx produced from the opening of CatSper channels is a necessary milestone in the process of fertilization and initiates hyperactivated motility (*Suarez, 2008*; *Carlson et al., 2005*, *2003*; *Chung et al., 2014*; *Jin et al., 2007*; *Navarro et al., 2008*; *Qi et al., 2007*; *Quill et al., 2003*; *Ren et al., 2001*).

CatSper channels exhibit weak voltage-dependency with half-maximal activation at $V_{1/2 \text{ human CatSper}}$ =+70 mV for capacitated sperm cells (*Lishko et al., 2011*). This parameter ($V_{1/2}$) reflects a certain membrane voltage condition under which 50% of CatSper channels are in the open state. Given this unusually high $V_{1/2}$, only a small fraction of human CatSper channels are open at physiological-relevant membrane potentials. P4 has been shown to potentiate CatSper activity by shifting $V_{1/2}$ to

*For correspondence:
lishko@berkeley.edu

Competing interests: The authors declare that no competing interests exist.

more negative values ($V_{1/2 \text{ human CatSper}}$ =+30 mV with 500 nM P4 for capacitated sperm cells [*Lishko et al., 2011*]). However, CatSper still requires both additional intracellular alkalization and significant membrane depolarization to function properly (*Lishko et al., 2011*; *Kirichok et al., 2006*). The proton channel Hv1 was revealed as one of the potential regulators of intracellular pH ($pH_i$) in human spermatozoa (*Lishko et al., 2010*; *Berger et al., 2017*). By mediating unidirectional flow of protons to the extracellular environment, Hv1 represents an important component in the CatSper activation cascade, but it also induces membrane hyperpolarization by exporting positive charges out of the cell. Hv1 is a voltage-gated channel and depends on membrane depolarization to be activated (*Ramsey et al., 2006*; *Sasaki et al., 2006*). Therefore, both CatSper and Hv1 must rely on yet unidentified depolarizing ion channels. P4 was shown to inhibit the $K^+$ channel of human sperm KSper ($IC_{50}$ = 7.5 ± 1.3 µM [*Mannowetz et al., 2013*; *Brenker et al., 2014*]) making KSper inhibition one of the potential origins for membrane depolarization. However, efficient KSper inhibition requires P4 concentrations in the µM range, which are only present in close vicinity of the egg. Sperm hyperactivation, however, occurs in the fallopian tubes, where P4 concentrations are not sufficient to block KSper (*Demott and Suarez, 1992*). Hence, the current model is missing a fourth member – the 'Depolarizing Channel of Sperm' (DSper) (*Miller et al., 2015*). Activation of the hypothetical DSper ion channel would induce long-lasting membrane depolarization and provide the necessary positive net charge influx for CatSper/Hv1 activity. Despite its central role, the molecular identity of DSper yet remains elusive.

The goal of this work was to characterize DSper and resolve its molecular identity in human spermatozoa. Using whole-cell voltage-clamp measurements, we recorded a novel non-CatSper conductance in both capacitated and noncapacitated spermatozoa. This unidentified, nonselective cation conductance exhibited outward rectification and pronounced temperature sensitivity in a range that matches the temperature spectrum of TRPV4 activity. Moreover, the pharmacological profile of DSper bears resemblance to TRPV4. Based on our electrophysiological, biochemical and immunocytochemical data, we thus conclude that the molecular identity of DSper is TRPV4.

## Results

### A novel non-CatSper conductance of human sperm cells

As many calcium channels, CatSper conducts monovalent ions, such as $Cs^+$ and $Na^+$ in the absence of divalent cations from the extracellular solution (divalent free; DVF) (*Lishko et al., 2011*; *Kirichok et al., 2006*). CatSper is also permeable to $Ca^{2+}$ and $Ba^{2+}$, but it cannot conduct $Mg^{2+}$ (*Figure 1—figure supplement 1*; *Figure 1—source data 1B*). In the presence of extracellular $Mg^{2+}$ the CatSper pore is blocked, resulting in the inhibition of monovalent CatSper currents ($I_{CatSper}$) (*Figure 1—figure supplement 1*; *Figure 1—source data 1B*).

In whole-cell voltage-clamp recordings from human ejaculated spermatozoa, we consistently observed residual currents when $I_{CatSper}$ was blocked with 1 mM extracellular $Mg^{2+}$ (*Figure 1A,B*). $Cs^+$ inward and outward currents elicited under DVF condition (black traces and bars) were larger than currents recorded in the presence of $Mg^{2+}$ (red traces and bars) (*Figure 1A–C*; *Figure 1—source data 1A*). This phenomenon was observed in both noncapacitated and capacitated spermatozoa, respectively. Notably, capacitated cells generally showed increased current densities under both conditions (*Figure 1C*). The data suggests that the remaining conductance is a novel non-CatSper conductance via the yet to be identified DSper ion channel. DSper currents were potentiated during capacitation (*Figure 1C*; *Figure 1—source data 1A*) and exhibited outward rectification, though, DSper currents recorded from capacitated cells were notably less rectifying (*Figure 1A,B*). This DSper component is unlikely a remnant of an increased leak current since the cells returned to their initial 'baseline' current after returning to the initial (HS) bath solution (*Figure 1—figure supplement 2*). Cation influx is the physiologically relevant entity to be analyzed as it represents channel activity under physiological relevant conditions and ensures membrane depolarization. Therefore, we preferentially analyzed DSper inward currents elicited by the change of membrane potential from 0 mV to −80 mV. To rule out 'contamination' of putative $I_{DSper}$ with remaining $I_{CatSper}$, we next tested whether 1 mM $Mg^{2+}$ is sufficient to completely block $I_{CatSper}$ and selectively isolate DSper currents. The CatSper inhibitor NNC 55–0396 (*Lishko et al., 2011*; *Strünker et al., 2011*) did not elicit any additional inhibitory effect on $I_{DSper}$ (*Figure 1D–F*; *Figure 1—source data 1A*), confirming

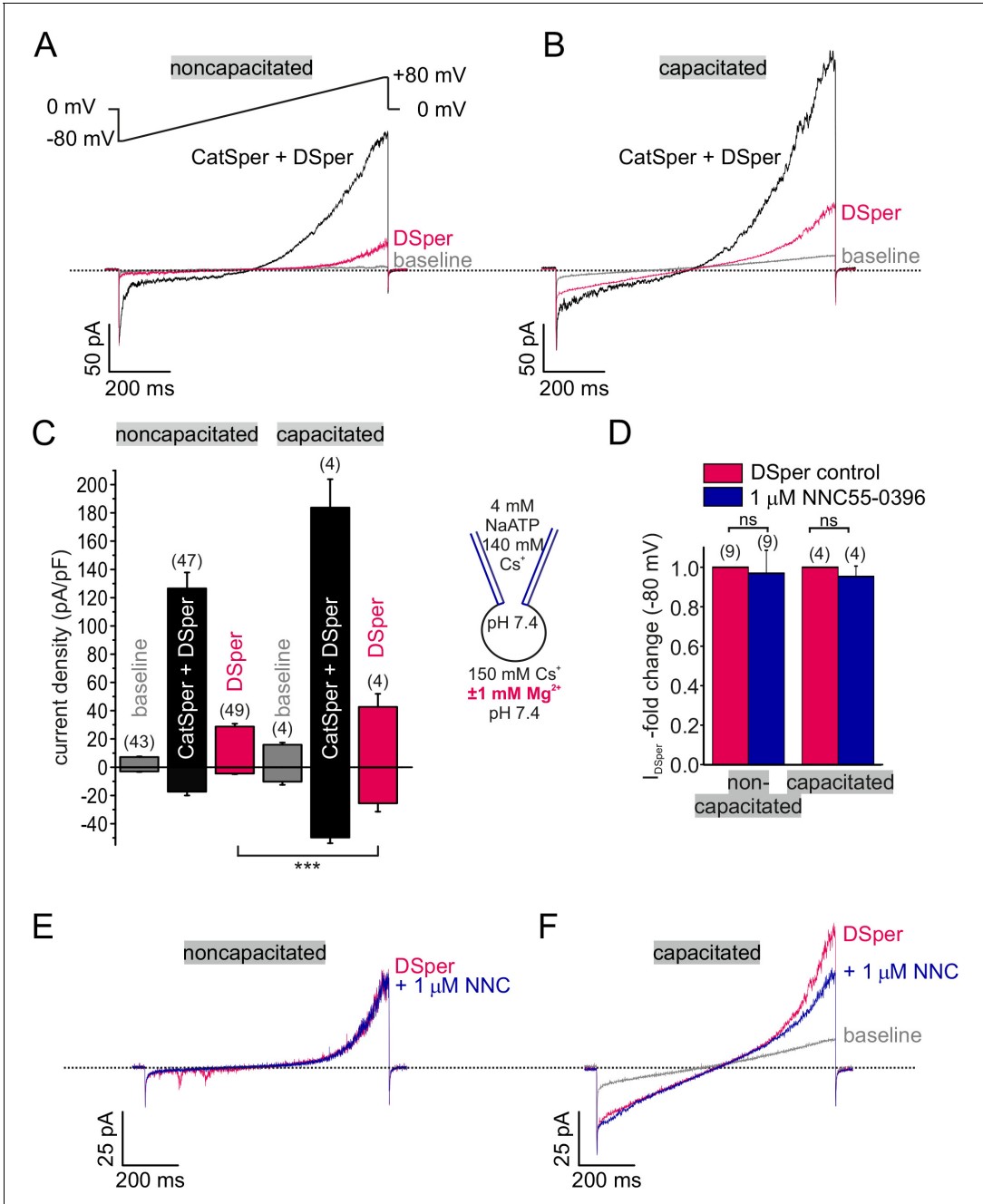

**Figure 1.** Electrophysiological recordings reveal a novel non-CatSper conductance. (**A-B**) Original current traces from representative whole-cell patch-clamp recordings from noncapacitated (**A**) and capacitated (**B**) human spermatozoa. Inward- and outward currents were elicited with voltage ramps as depicted in (**A**). Under divalent free conditions (black traces), typical CatSper monovalent caesium currents can be recorded. In the presence of 1 mM $Mg^{2+}$ (red traces), an outward rectifying 'DSper' current component remains. Hence, the black traces represent a mixture of both CatSper and DSper monovalent $Cs^+$ currents, while the red traces show pure $Cs^+$ currents through DSper. (**C**) Quantification of current densities for all three conditions in (**A-B**). DSper currents are potentiated upon capacitation (noncapacitated cells: -4.50 ± 0.41 pA/pF, capacitated cells: -25.58 ± 5.88 pA/pF for inward currents recorded at -80 mV; noncapacitated cells: 28.80 ± 1.93 pA-pF, capacitated cells: 42.67 ± 9.27 pA/pF for outward currents recorded at +80 mV). Data are mean ± S.E.M., with (n) representing the number of individual sperm cells tested. Statistical significance (unpaired t-test) was indicated by: ***p ≤0.0005. Data was collected from 3 human donors, no variations between human donors were noticed. Quantification of normalized DSper inward currents (**D**) and original current traces (**E-F**) in presence and absence of the CatSper inhibitor NNC 55-0396 demonstrate the absence of the inhibition.
DOI: https://doi.org/10.7554/eLife.35853.002

The following source data and figure supplements are available for figure 1:

**Source data 1.** The source data for the inward and outward monovalent and divalent DSper current densities.

*Figure 1 continued on next page*

*Figure 1 continued*

DOI: https://doi.org/10.7554/eLife.35853.005

**Figure supplement 1.** Human CatSper conducts $Ca^{2+}$ and $Ba^{2+}$ but not $Mg^{2+}$.

DOI: https://doi.org/10.7554/eLife.35853.003

**Figure supplement 2.** DSper currents are recorded under stable conditions.

DOI: https://doi.org/10.7554/eLife.35853.004

efficient CatSper pore block by $Mg^{2+}$. These findings corroborate our hypothesis that a novel CatSper-independent cation conductance could provide additional depolarization under physiological conditions. To isolate $I_{DSper}$, we performed all following experiments in presence of both $Mg^{2+}$ and NNC 55–0396.

## Human sperm DSper current exhibits temperature sensitivity

We next aimed to investigate mechanism(s) of DSper activation. Previous work had focused on various DSper candidates, one being ATP-activated P2X channels. Navarro *et al.* showed functional expression of P2X2 in mouse spermatozoa (*Navarro et al., 2011*). However, human spermatozoa appear to be insensitive to extracellular ATP (*Brenker et al., 2012*). De Toni *et al.* suggested that human spermatozoa perform thermotaxis mediated by a member of the thermosensitive transient receptor potential vanilloid channel family, TRPV1 (*De Toni et al., 2016*), supporting their claim by immunocytochemistry and $Ca^{2+}$ imaging. By contrast, Kumar *et al.* detected TRPV4 expression in human spermatozoa using immunocytochemistry and calcium imaging (*Kumar et al., 2016*). To date, several temperature-sensitive ion channels and specific transporters have been reported in mammalian sperm (*Kumar et al., 2016*; *Gervasi et al., 2011*; *Hamano et al., 2016*). However, functional characterization of a temperature-activated cation conductance via direct methods, such as electrophysiology, has not been performed in human sperm yet. Since the functional expression of a thermosensitive TRP ion channel in human spermatozoa is currently under debate, and their cation permeability renders many of them DSper candidates, we investigated the impact of temperature on DSper activity. As shown in *Figure 2A–C*, elevating temperature profoundly increased $I_{DSper}$. We observed a temperature-induced potentiation of both inward and outward currents in noncapacitated, as well as capacitated human spermatozoa (*Figure 2A–B*; *Figure 2—source data 1*). A temperature ramp from 23°C to 37°C potentiated $I_{DSper}$ inward currents by factors of 2.7 $\pm$ 0.5 for noncapacitated cells and 2.0 $\pm$ 0.2 for capacitated cells, respectively ($Q_{10\ noncapacitated}$=1.76, $Q_{10\ capacitated}$=1.65 for caesium inward currents). Half-maximal activation was achieved at $T_{1/2}$ = 34°C (noncapacitated) and $T_{1/2}$ = 31°C (capacitated) (*Figure 2D*). Moreover, the temperature-induced potentiation effect was reversible for both noncapacitated and capacitated cells (*Figure 2E*). We hence concluded that the observed phenomenon is not a temperature-induced loss of the seal and compromised membrane stability and that DSper is indeed temperature-activated.

## DSper conducts sodium ions

Since sodium ($Na^+$) is the major extracellular ion in the female reproductive tract ($[Na^+]$=140–150 mM [*Borland et al., 1980*]), $Na^+$ is a likely source for membrane depolarization. We therefore investigated whether DSper has the capacity to conduct $Na^+$. As indicated in *Figure 3A*, similar outward rectifying DSper currentswere recorded when extracellular $Cs^+$ was replaced with equimolar concentrations of $Na^+$. DSper inward $Na^+$ currents were entirely CatSper-independent, since NNC 55–0396 had no significant inhibitory effect (*Figure 3B,C*; *Figure 3—source data 1*). In the presence of both 1 mM $Mg^{2+}$ and 1 µM NNC 55–0396, $I_{DSper}$ was still reversibly activated by warm temperatures (*Figure 3D,E*; *Figure 3—source data 1*) with a 4.1 $\pm$ 0.5 fold increase for the inward sodium currents from 22°C to 37°C, which is notably larger than the fold-increase as observed for cesium currents (*Figure 2D*). Half -maximum activation was at $T_{1/2}$ = 34°C, comparable to previously analyzed values for the temperature-activated $Cs^+$ currents, however sodium conductance via DSper produced a larger $Q_{10\ noncapacitated}$=2.30. Together, these electrophysiological data indicate that DSper shares characteristic hallmarks with thermosensitive TRPV channels (*Benham et al., 2003*). We thus proceeded to define which TRPV channel(s) is involved.

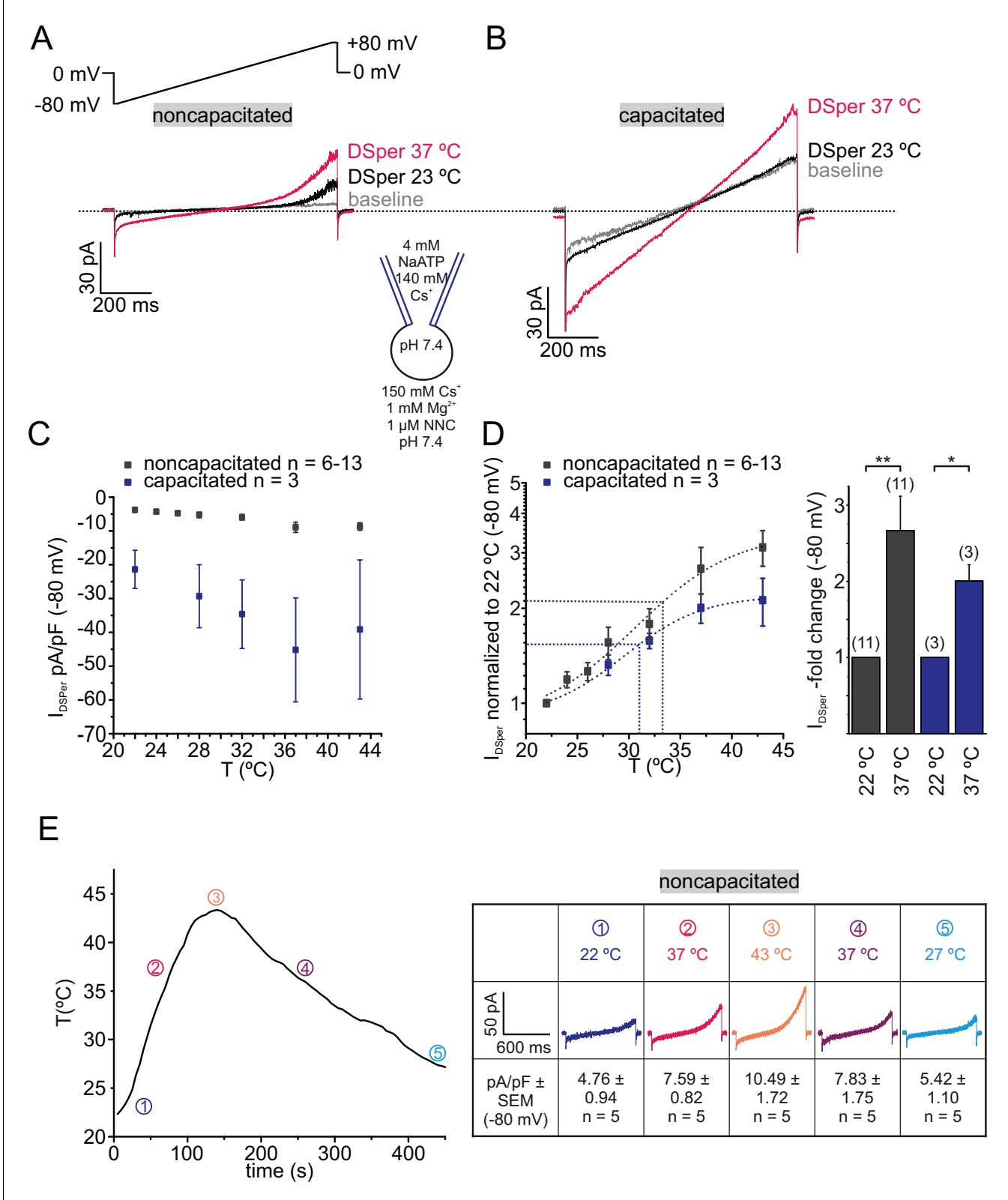

**Figure 2.** DSper is activated by warm temperatures. (A–B) Representative current traces from whole-cell patch-clamp recordings from noncapacitated (A) and capacitated (B) human spermatozoa challenged with a rise in temperature from 23°C to 37°C. Both DSper inward- and outward currents are increased at warmer temperatures. (C) Quantification of DSper inward current densities as a function of bath temperature (in °C). Noncapacitated (grey squares) as well as capacitated cells (blue squares) show increased current densities when stimulated with increasing bath temperatures. (D) Data of (C)

*Figure 2 continued on next page*

*Figure 2 continued*

normalized to room temperature (22°C). Half maximal activation at $T_{1/2}$ = 34°C (noncapacitated) and $T_{1/2}$ = 31°C (capacitated) indicated by the dotted lines. The data were fitted with Boltzmann equation to estimate the temperature at which DSper currents have half-maximal activation. Right panel: statistical significance (unpaired t-test) was indicated by: *p≤0.05, **p≤0.005 for capacitated (blue bars) and noncapcited (black bars) human sperm. (E) The bath temperatures as a function of time and corresponding DSper currents. Inset shows representative traces indicating that the temperature-induced potentiation effect was reversible. Data are mean ± S.E.M., with (n) representing the number of individual sperm cells tested obtained from three human donors.

DOI: https://doi.org/10.7554/eLife.35853.006

The following source data is available for figure 2:

**Source data 1.** DSper inward current densities as a function of the bath temperature.

DOI: https://doi.org/10.7554/eLife.35853.007

## DSper is represented by the cation channel TRPV4

Based on the observed $I_{DSper}$ temperature spectrum (*Figures 2D* and *3E*), candidate channels could be TRPV3, TRPM3 or TRPV4 (*Benham et al., 2003*; *Güler et al., 2002*; *Watanabe et al., 2002*; *Cheng et al., 2012*). We have ruled out TRPV2 involvement, since TRPV2 has an unusually steep activation threshold of above 53°C (*Moore and Liedtke, 2017*). In addition, TRPV1 was previously proposed as a mediator of human sperm thermotaxis (*De Toni et al., 2016*). To discriminate between these channels, we tested potential effects of corresponding selective agonists – carvacrol (*Vogt-Eisele et al., 2009*) for TRPV3, RN1747 (*Vincent et al., 2009*) for TRPV4, capsaicin (*Caterina et al., 1997*) for TRPV1, and pregnenolone sulfate (*Harteneck, 2013*) for TRPM3. Employing either electrophysiological, or $Ca^{2+}$ imaging recordings, only TRPV4 agonist RN1747 elicited a significant effect. In detail, application of 10 μM RN1747 ($EC_{50}$ = 0.77 μM [*Vincent et al., 2009*]) significantly potentiated DSper outward currents (*Figure 4A,B*; *Figure 4—source data 1A*) in noncapacitated human sperm. In contrast, no effects were observed by 1 μM or 10 μM capsaicin ($EC_{50}$ = 711.9 nM [*Caterina et al., 1997*]) (*Figure 4—figure supplement 1*, *Figure 4—source data 1B-D*). In order to confirm TRPV1 functional absence, we repeated these capsaicin experiments with 30 μM $PI_{4,5}P_2$ inside, to account for a possible loss of capsaicin sensitivity due to potential depletion of endogenous $PI_{4,5}P_2$ during whole-cell recording (*Bevan et al., 2014*; *Senning et al., 2014*). However, no change in DSper inward- and outward currents was observed. Using $Ca^{2+}$ imaging of fluo-4/AM-loaded sperm, we next recorded fluorescence changes in the flagellar principle piece while stimulating human sperm with either 10 μM capsaicin or 500 μM carvacrol (*Figure 4—figure supplement 1*; *Figure 4—source data 1C,D*). Neither TRPV1 nor TRPV3 agonist elicited any rise in cytosolic calcium levels. We thus concluded that human spermatozoa do not express functional TRPV1 or TRPV3 channels. TRPM3 channels also exhibit temperature sensitivity between ambient warm to hot, which resembles the range observed for DSper (*Vriens and Voets, 2018*). Therefore, we have tested the possibility of TRPM3 involvement in $I_{DSper}$ generation by applying the TRPM3 agonist pregnenolone sulfate (PS) (*Figure 4—figure supplement 1E*; *Figure 4—source data 1E*). Application of 10 μM PS did not result in any change of the basal DSper current, confirming the absence of functional TRPM3 in human spermatozoa. Taking together, our results indicate that the temperature-activated cation channel TRPV4 is likely to be functionally expressed and provides membrane depolarization in human sperm.

Interestingly, additional pharmacological investigation of DSper revealed that both TRPV4-specific antagonists, HC067047 and RN1734 (*Vincent et al., 2009*; *Everaerts et al., 2010*), prevented temperature activation of DSper, confirming that DSper pharmacology matches that of TRPV4 (*Figure 5A–C*; *Figure 5—source data 1A*). Since both inhibitors are dissolved in ethanol, we performed a vehicle control to exclude any inhibitory effect of ethanol on temperature activation. Indeed, the same vehicle concentration (0.1% ethanol) failed to inhibit DSper temperature activation and yielded results comparable to the control conditions (*Figure 5—figure supplement 1*; *Figure 5—source data 1B*).

Supporting our functional data, TRPV4 was detected in human sperm on both mRNA and protein levels. Reverse transcriptase PCR performed with mRNA isolated from "swim-up" purified spermatozoa, followed by an amplification of the full-length TRPV4 (*Figure 5—figure supplement 2A*), produced a band of the expected size. The band was absent in negative controls, to which no reverse

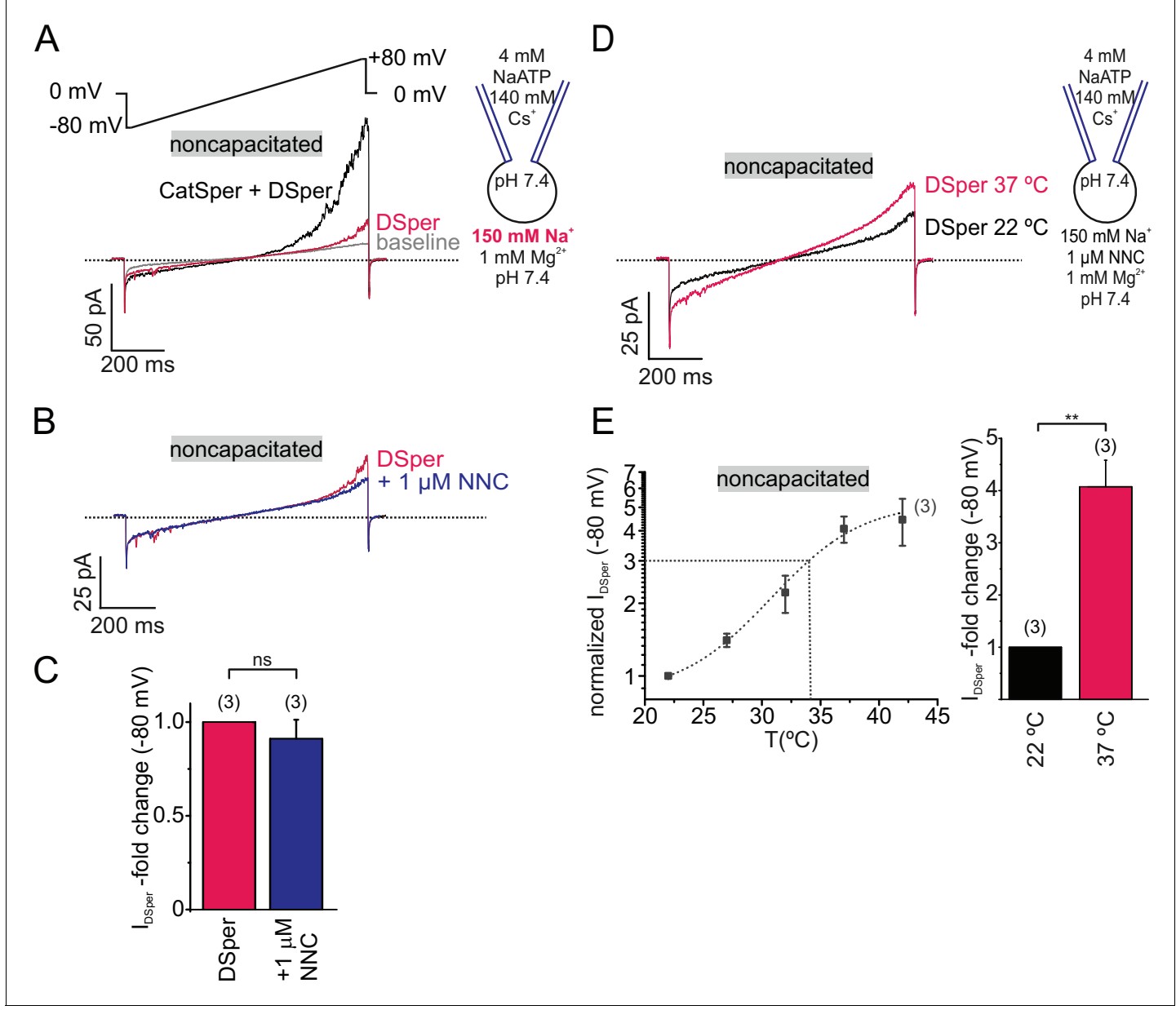

**Figure 3.** DSper conducts sodium ions. (A) Representative current traces from the whole-cell patch-clamp recordings of noncapacitated human spermatozoa. Inward and outward currents were elicited with the voltage ramps as depicted. To record DSper currents, extracellular $Cs^+$ was substituted with the same concentration of $Na^+$. Representative current traces (B) and quantification of the normalized DSper inward currents (C) before and after stimulation with 1 µM NNC suggest that CatSper channel does not contribute to the recorded sodium inward conductance. (D–E) Representative current traces in (D) and quantification of the inward currents normalized to 22°C (E) at increasing bath temperatures. A similar temperature-induced potentiation effect of DSper sodium inward currents was observed as for caesium currents. Recordings were performed in the presence of NNC to exclude any CatSper contribution. Half maximal activation was achieved at $T_{1/2}$sodium = 34°C (dotted line). The data were fitted with the Boltzmann equation. Statistical significance (unpaired t-test) is indicated by: **$p \leq 0.005$. Data are mean ± S.E.M., with (n) representing the number of individual sperm cells tested obtained from two human donors.

DOI: https://doi.org/10.7554/eLife.35853.008

The following source data is available for figure 3:

**Source data 1.** The source data for the inward sodium currents via DSper.

DOI: https://doi.org/10.7554/eLife.35853.009

transcriptase and no templates were added. The sequence of the isolated PCR product of that specific band (dotted square), yielded the full-length sequence of TRPV4 isoform A (2620 bp, 98 kDa,

Q9ERZ8). Moreover, the presence of TRPV4 protein was confirmed by western blotting (*Figure 5—figure supplement 2B*). Immunoreactive bands were detected at ~115 kDa in extracts from human testicular tissue (1), capacitated (2) and noncapacitated (3) spermatozoa (*Figure 5—figure supplement 2B*). Immunostaining with anti-hTRPV4 specific antibodies (*Figure 5—figure supplement 2C*) yielded an immunopositive signal in the acrosome and flagellum. Finally, when TRPV4 was cloned from human sperm mRNA extracts and recombinantly expressed in HEK293 cells (*Figure 6A*), a band of similar molecular weight could be detected by western blotting (*Figure 6B*). Moreover, TRPV4 cloned from human sperm mRNA recapitulates DSper temperature sensitivity (*Figure 6D–E*; *Figure 6—source data 1*), as well as activation by the selective TRPV4 agonist RN1747 (*Figure 6F–G*; *Figure 6—source data 1*), indicating that TRPV4 cloned from human sperm cells indeed assembles into a functional channel.

## Discussion

Sperm transition to hyperactivated motility is essential for fertilization. Hyperactivation provides the propulsion force required to penetrate through viscous luminal fluids of the female reproductive tract and protective vestments of the egg. The CatSper channel is a key player in the transition to hyperactivated motility (*Ren et al., 2001*). However, proper CatSper function requires three concurrent activation mechanisms: (1) membrane depolarization (*Lishko et al., 2011*; *Kirichok et al., 2006*), (2) intracellular alkalization (*Lishko et al., 2011*; *Kirichok et al., 2006*), and for primate CatSper specifically (3) abundance of progesterone (*Lishko et al., 2011*; *Strünker et al., 2011*; *Smith et al., 2013*; *Sumigama et al., 2015*). While the two latter mechanisms have been described in detail, the source of membrane depolarization remained puzzling.

In human spermatozoa, K$^+$, Ca$^{2+}$, Cl$^-$, and H$^+$ conductances have been described (*Lishko et al., 2011*; *Strünker et al., 2011*; *Smith et al., 2013*; *Lishko et al., 2010*; *Berger et al., 2017*; *Mannowetz et al., 2013*; *Brenker et al., 2014*; *Brown et al., 2016*; *Geng et al., 2017*; *Williams et al., 2015*; *Orta et al., 2012*). However, the molecular nature of Na$^+$ conductance of sperm remained unknown. Upon ejaculation, mammalian spermatozoa are exposed to increased

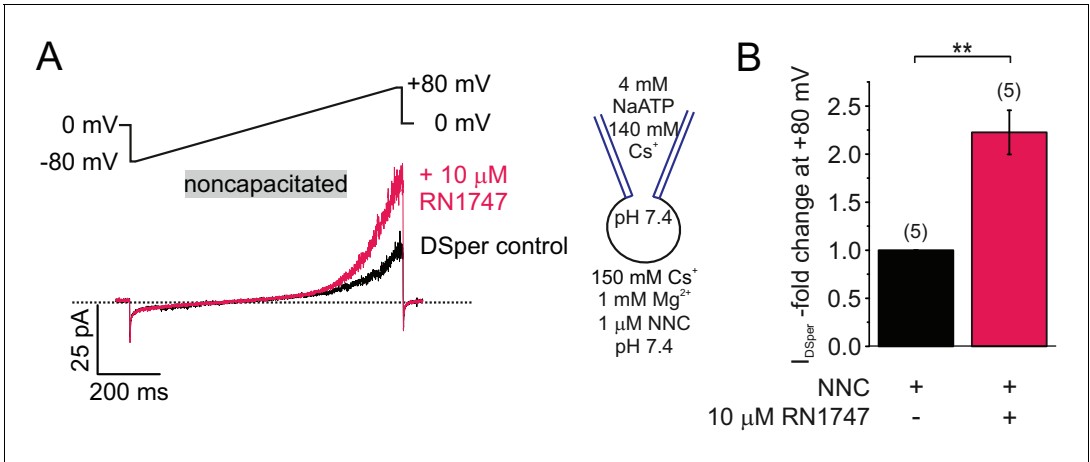

**Figure 4.** DSper is activated by the TRPV4 agonist RN1747. (**A**) Representative whole-cell patch-clamp recordings of noncapacitated human spermatozoa. Inward- and outward currents were elicited with voltage ramps as depicted. DSper monovalent caesium currents (black trace) are increased after stimulation with 10 µM RN1747 (red trace). Both recordings were performed in the presence of 1 µM NNC. (**B**) Quantification of normalized DSper outward currents under control conditions and after stimulation with RN1747. A significant gain upon stimulation with the TRPV4 agonist (factor 2.22 ± 0.23, **p=0.0007, unpaired t-test, n = 6) is observable. No variation between human donors were noticed.

DOI: https://doi.org/10.7554/eLife.35853.010

The following source data and figure supplement are available for figure 4:

**Source data 1.** The source data for DSper regulation of TRP channel agonists and inhibitors.

DOI: https://doi.org/10.7554/eLife.35853.012

**Figure supplement 1.** TRPV1, TRPV3 and TRPM3 are not functionally expressed in human spermatozoa.

DOI: https://doi.org/10.7554/eLife.35853.011

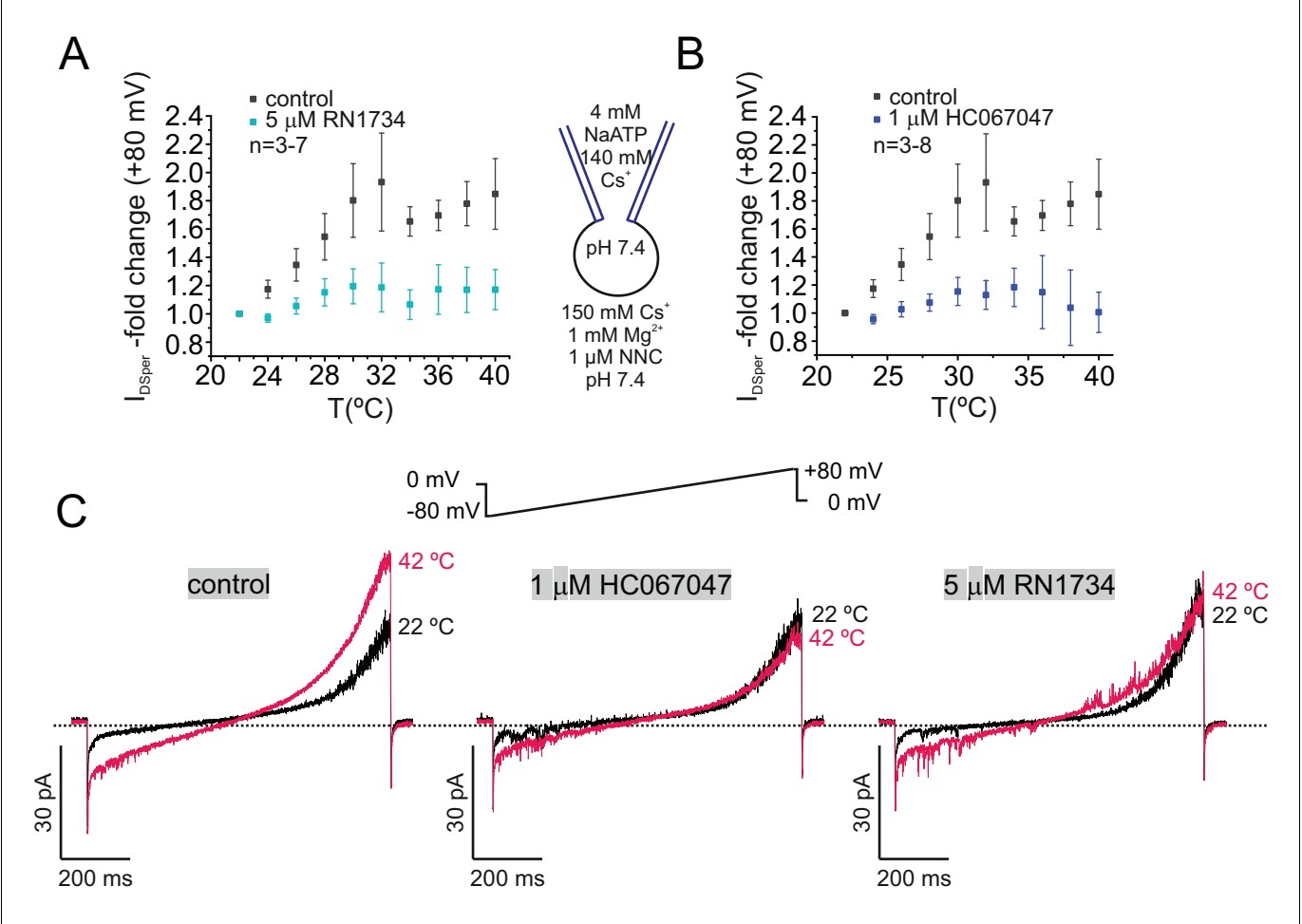

**Figure 5.** TRPV4 inhibitors RN1734 and HC067047 antagonize human DSper activity. (A–B) Quantification of normalized DSper outward currents as a function of bath temperature (in °C) in the absence (grey squares) and the presence (turquoise/blue squares) of TRPV4 inhibitors RN1734 (A) and HC067047 (B), respectively. Both inhibitors greatly reduce DSper's temperature-sensitivity. Representative current traces of whole-cell voltage clamp recordings from noncapacitated spermatozoa for all three conditions are depicted in (C). Representative current traces indicate that a rise in temperature from 22°C to 42°C considerably potentiated DSper inward and outward currents under control conditions but not in presence of the TRPV4 antagonists HC067047 and RN1734. No variations between human donors were noticed.

DOI: https://doi.org/10.7554/eLife.35853.013

The following source data and figure supplements are available for figure 5:

**Source data 1.** The source data for the experiments shown on *Figure 5* and its *Figure 5—figure supplement 1*.
DOI: https://doi.org/10.7554/eLife.35853.016

**Figure supplement 1.** Ethanol does not affect DSper temperature activation.
DOI: https://doi.org/10.7554/eLife.35853.014

**Figure supplement 2.** TRPV4 can be detected on the protein and mRNA levels.
DOI: https://doi.org/10.7554/eLife.35853.015

[Na$^+$] (~30 mM in cauda epididymis *versus* 100–150 mM in seminal plasma). In the female reproductive tract, Na$^+$ levels are similar to those in serum (140–150 mM) (*Borland et al., 1980*). Hence, Na$^+$ is ideally suited to provide a depolarizing charge upon sperm deposit into the female reproductive tract.

Here, we recorded a novel CatSper-independent cation conductance that exhibits outward rectification as well as potentiation upon capacitation. We propose that this novel conductance is carried by the hypothetical 'Depolarizing Channel of Sperm' DSper and provides the necessary cation influx that ensures membrane depolarization. $I_{DSper}$ is activated by warm temperatures between 22 and 37°C (*Figures 2* and *3D–E*) which makes the protein thermoresponsive within the physiologically

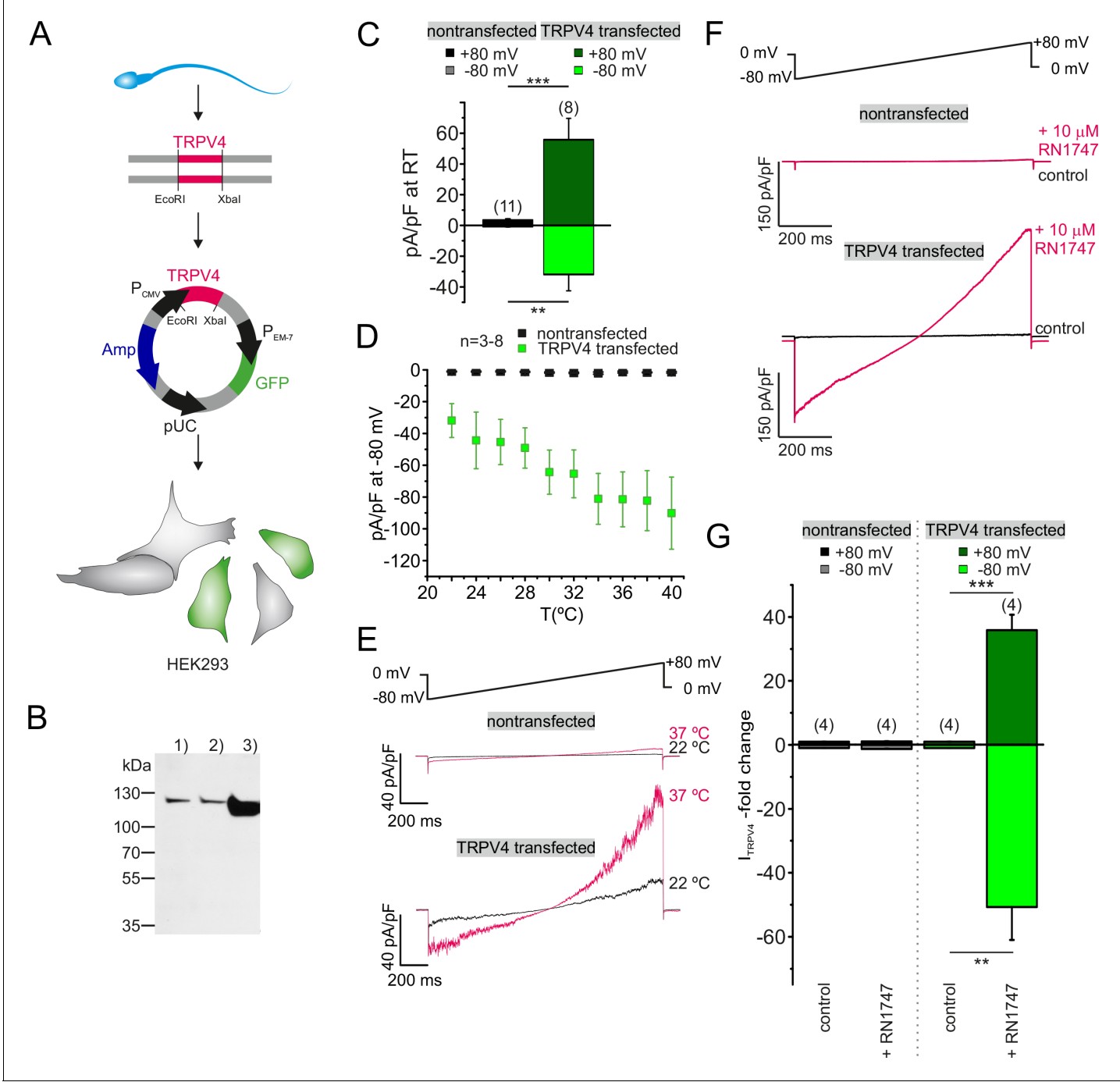

**Figure 6.** Sperm TRPV4 assembles into a functional channel when recombinantly expressed in HEK293 cells. (**A**) Schematic representation of experimental approach: Full-length TRPV4 (isoform A) was cloned from human sperm mRNA extracts and recombinantly expressed in HEK293 cells, utilizing a pTracer CMV vector (Invitrogen). The GFP containing bicistronic vector allowed identification of green fluorescent transfected cells. (**B**) Western blotting results are shown for (1) nontransfected HEK293 cells, (2) cells transfected with the empty vector and (3) HEK293 cells transfected with the TRPV4-containing vector. An intense immunopositive band can be detected in line 3) at approx. 115 kDa corresponding to the expected size for TRPV4. Weak bands in 1) and 2) suggest endogenous expression of TRPV4 in HEK293 cells. (**C**) Electrophysiological characterization reveals a significant increase in the basic activity of TRPV4 transfected cells (at room temperature, 22°C). Inward and outward current densities were recorded as described before (ramp recordings −80 to +80 mV). Statistical significance (unpaired t-test) is indicated by: **p≤0.005, ***p≤0.0005. (**D**) Quantification of the inward current densities for nontransfected vs. TRPV4 transfected cells as a function of bath temperature (in °C) and representative current traces of whole-cell voltage clamp recordings for both cell populations in (**E**). Temperature-induced potentiation of inward and outward currents in TRPV4 transfected cells suggests that TRPV4 cloned from human sperm assembles into a functional protein in the heterologous expression system. (**F**) Whole-
*Figure 6 continued on next page*

*Figure 6 continued*

cell voltage clamp recordings reveal a strong potentiation of the inward and outward currents upon stimulation with the TRPV4 agonist RN1747 in TRPV4- transfected but not in nontransfected cells. Inward and outward currents were elicited via voltage ramps as depicted above. (**G**) Quantification of both inward and outward currents normalized to control conditions indicates statistical significance (**p≤0.005, ***p≤0.0005) for TRPV4 transfected cells.

DOI: https://doi.org/10.7554/eLife.35853.017

The following source data is available for figure 6:

**Source data 1.** Recombinantly expressed TRPV4, initially isolated from human sperm mRNA pool, exhibits typical TRPV4-like behavior when expressed in HEK293 cells.

DOI: https://doi.org/10.7554/eLife.35853.018

relevant temperatures (34.4°C in the epididymis (*Valeri et al., 1993*), 37°C body core temperature at the site of fertilization). Previous studies showed that capacitated rabbit and human sperm cells have an inherent temperature sensing ability (*Bahat et al., 2003*), which could be an additional driving force to guide male gametes from the reservoir in the Fallopian tubes towards the warmer fertilization site. It is thus very likely, that human spermatozoa express a temperature-activated ion channel, which operates in the described temperature range and enables thermotaxis. Another potential role for this channel is to serve as a sensor for the initiation of human sperm capacitation. During maturation in the female reproductive tract, human sperm are exposed to elevated temperature, especially before and during ovulation, which is correlated with an increase in basal body temperature by 1°C. As spermatozoa are able to survive in the female reproductive tract for several days by binding to the ciliated epithelia of the fallopian tubes, they must eventually undergo hyperactivation to detach (*Ardon et al., 2016*). Accordingly, CatSper-deficient spermatozoa that cannot hyperactivate are not able to ascend the fallopian tubes (*Ho et al., 2009*).

In order for hyperactivation to occur, spermatozoa must be fully capacitated – a process that takes approximately 5 hr in humans, and requires sperm exposure to bicarbonate, albumin, and elevated temperature. While sperm capacitation can be achieved in vitro, exposure to 37°C is an absolute requirement. Therefore, the presence of a temperature-sensitive sperm ion channel could serve as the potential sensor for the onset of capacitation and might ensure sperm final maturation in the female reproductive tract.

The temperature response profile of DSper conforms with previously reported temperature sensitivity of TRPV4 (*Benham et al., 2003*; *Güler et al., 2002*; *Watanabe et al., 2002*). Moreover, we observed $I_{DSper}$ potentiation by the selective TRPV4 agonist RN1747 (*Figure 4*), as well as decreased temperature-sensitivity upon stimulation with TRPV4 selective inhibitors HC067047 and RN1734 (*Figure 5*). It should be noted that the absence of a significant inward current via TRPV4 in presence of both extracellular $Mg^{2+}$ and $Ca^{2+}$, such as in HS solution (*Figure 1A–B*), results from competition for the channel pore between the divalent and monovalent ions. It has been reported that extracellular $Ca^{2+}$ inhibits TRPV4 monovalent conductance (*Watanabe et al., 2003*).

The temperature coefficient $Q_{10}$ reflects the temperature dependence of the membrane current and has been reported to be between 9 and 19 for recombinantly expressed TRPV4 (*Güler et al., 2002*; *Watanabe et al., 2002*). However, endogenously expressed TRPV4 channels recorded from aorta endothelial cells (*Watanabe et al., 2002*) exhibit less steep temperature dependence, which resembles the $Q_{10}$ of sperm TRPV4 ($Q_{10 \, sodium}$ = 2.30, noncapacitated sperm). It is possible that different lipid environments or additional channel modifications are responsible for such differences.

It is also possible that sperm cells possess more than one type of temperature-regulated ion channel. The biphasic inhibition of DSper with TRPV4-selective antagonists (*Figure 5*) does not result in complete current inhibition, particularly in the temperate range between 24 and 32°C. This may suggest an additional, non-TRPV4 conductance. The molecular nature of such additional conductance(s) could be either temperature-dependent release of NNC inhibition on CatSper or perhaps the presence of other yet undiscovered temperature-sensitive ion channel(s). Interestingly, according to one published report (*Hamano et al., 2016*), murine TRPV4 regulates sperm thermotaxis. However, TRPV4-deficient male mice are fertile which may indicate either presence of an additional temperature sensor or a compensatory mechanism.

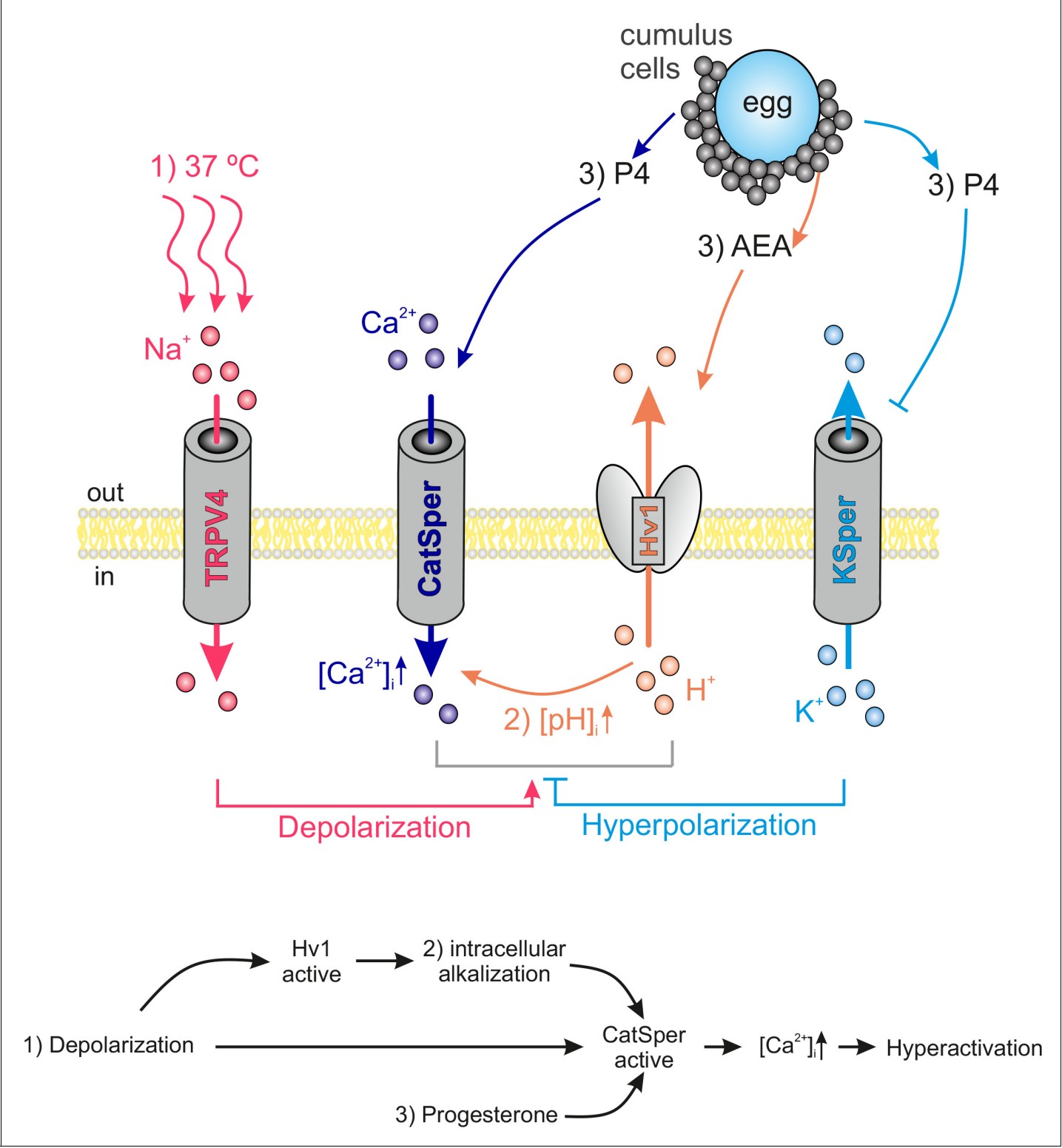

**Figure 7.** Interdependency of ion channel complexes in the human sperm flagellum. Transition into hyperactivated motility is triggered by a CatSper-mediated rise in the cytosolic calcium levels. Proper CatSper function requires three concurrent activation mechanisms: (1) membrane depolarization, (2) intracellular alkalization via Hv1-mediated proton extrusion, and (3) abundance of progesterone. In our proposed model the sperm's sodium channel TRPV4 is activated by warm temperatures (37°C at the site of fertilization). TRPV4-mediated sodium influx induces: (1) membrane depolarization, which in turn activates both Hv1 and CatSper. Hv1 then extrudes protons out of the sperm, thereby leading to (2) intracellular alkalization and further activation of CatSper. Cumulus cells surrounding the egg secrete (3) P4 and AEA. P4 releases CatSper inhibition and high P4 concentrations block

*Figure 7 continued on next page*

*Figure 7 continued*

KSper-mediated hyperpolarization. AEA was shown to potentiate Hv1. The resulting opening of CatSper generates a Ca$^{2+}$ influx that serves as the trigger for hyperactivation.

DOI: https://doi.org/10.7554/eLife.35853.019

According to our model (*Figure 7*), human spermatozoa are exposed to an increase in both temperature and [Na$^+$] upon deposit to the female reproductive tract. TRPV4-mediated Na$^+$ influx induces membrane depolarization, which in turn activates both Hv1 and CatSper. H$^+$ efflux through Hv1 promotes intracellular alkalization and thus enhanced CatSper activation. Approaching the egg, sperm is exposed to P4 and the endocannabinoid anandamide (AEA), both secreted by cumulus cells (*El-Talatini et al., 2009*; *Hunter and Rodriguez-Martinez, 2004*). P4 binding to ABHD2 releases CatSper inhibition (*Miller et al., 2016*) while AEA was shown to activate Hv1 (*Lishko et al., 2010*). The resulting opening of CatSper generates a Ca$^{2+}$ influx along the flagellum and serves as the trigger for hyperactivation. P4 not only potentiates CatSper, it also inhibits KSper-mediated hyperpolarization, which gives the CatSper activation cascade an additional impulse (*Mannowetz et al., 2013*).

Using a CatSper2-deficient infertile patient, no remaining cation current was recordable, when both Hv1 and KSper were blocked (*Smith et al., 2013*). However, these recordings were performed in a condition where ATP was absent from the pipette solution. According to Phelps *et al.* intracellular ATP binding to the N-terminal ankyrin repeat domain of TRPV4 has a profound sensitizing effect (*Phelps et al., 2010*; *Lishko et al., 2007*), which is a feature of the TRPV ankyrin repeats and is shared between TRPV1 and TRPV4 (*Lishko et al., 2007*). Indeed, addition of 4 mM ATP to the pipette solution, allowed us to consistently record TRPV4 activity from fertile human sperm.

Our data suggests that TRPV4 activity is increased upon capacitation. Since capacitation encompasses changes in the phosphorylation state of many proteins (*Visconti et al., 2011*), and TRPV4 requires tyrosine phosphorylation to function properly (*Wegierski et al., 2009*), it is likely that TRPV4 phosphorylation is required. It would also explain, why only capacitated human spermatozoa appear to be thermotactically responsive (*Bahat et al., 2003*). Interestingly, we also observed different $I_{DSper}$ kinetics (i.e. less outward rectification) after capacitation. This finding could also be the result of phosphorylation, modified lipid composition or even formation of TRPV4/X heteromers upon capacitation. These aspects will be addressed in future studies.

Selective anti-hTRPV4 antibodies located TRPV4 in the flagellum and acrosome of human sperm (*Figure 5—figure supplement 2C*). The localization of TRPV4 in the acrosome region should be evaluated critically since this compartment is highly antigenic and attracts antibodies in general (*Cheng et al., 1996*). However, TRPV4 appears to be distributed in the sperm flagellum. The principal piece of the sperm tail is also the compartment where CatSper and Hv1 reside (*Lishko et al., 2010*; *Ren et al., 2001*), bringing those three interdependent ion channels in close proximity to each other.

TRPV4 – more precisely its hyperfunction - might underlie the aversive effect of increased scrotal temperatures on sperm production and epididymal preservation. As proposed by Bedford *et al.*, increased scrotal temperatures when clothed contribute substantially to the inferior quality of human ejaculate (*Bedford, 1991*). By contrast, TRPV4 might represent an attractive target for male fertility control, since TRPV4 is likely to lie upstream in the signaling cascades leading to sperm hyperactivation and can be heterologously expressed for high-throughput functional studies.

# Materials and methods

**Key resources table**

| Reagent type | Designation | Source | Identifier | Additional information |
|---|---|---|---|---|
| Cell line | HEK293 | ATCC | ATCC cat# CRL-1573 RRID:CVRL_0045 | |
| Compound | RN1747 | Tocris | 3745 | TRPV4 agonist |

*Continued on next page*

*Continued*

| Reagent type | Designation | Source | Identifier | Additional information |
|---|---|---|---|---|
| Compound | capsaicin | Cayman | 404-86-4 | TRPV1 agonist |
| Compound | carvacrol | Sigma | 499-75-2 | TRPV3 agonist |
| Compound | Pregnenolone sulfate | Sigma | P162 | TRPM3 agonist |
| Compound | HC067047 | Tocris | 4100 | TRPV4 antagonist |
| Compound | RN1734 | Tocris | 3746 | TRPV4 antagonist |
| Compound | NNC 55–0396 | R and D systems | 2268 | CatSper antagonist |
| Compound | Fluo-4/AM | Invitrogen | F14201 | $Ca^{2+}$ indicator |
| Compound | Lipofectamine | Invitrogen | 11668019 | Transfection reagent |
| Antibody | α-TRPV4 | Alomone | Alomone cat# ACC-034 RRID:AB_2040264 | polyclonal; host: rabbit; unconjugated |

## Human sperm cells

A total of 5 healthy male volunteers were recruited to this study, which was conducted with approval of the Committee on Human Research at the University of California, Berkeley (protocol 10–01747, IRB reliance #151). Informed consent was obtained from all participants. Ejaculates were obtained by masturbation and spermatozoa were purified following the swim-up protocol as previously described (*Lishko et al., 2011*). In-vitro capacitation was accomplished by 4 hr incubation in 20% Fetal bovine serum, 25 mM $NaHCO_3$ in HTF buffer (*Lishko et al., 2013*) at 37°C and 5% $CO_2$.

## Reagents

RN1747, HC067047 and RN1734 were purchased from Tocris Bioscience (Bristol, UK), NNC 55–0396 was purchased from R&D systems (Minneapolis, USA), capsaicin from Cayman Chemical (Ann Arbor, USA), fluo-4/AM is from Invitrogen (Thermo Fisher Scientific, Carlsbad, USA) and all other compounds were obtained from Sigma (St. Louis, USA). NNC 55–0396 stock solution was dissolved in water, HC067047 and RN1734 were dissolved in EtOH, while RN1747 and PS were dissolved in DMSO.

## Electrophysiology

For electrophysiological recordings, only the ultra-pure upper 1 ml of the swim-up fraction was used. Single cells were visualized with an inverse microscope (Olympus IX71) equipped with a differential interference contrast, a 60 x Objective (Olympus UPlanSApo, water immersion, 1.2 NA, ∞/0.13–0.21/FN26.5) and a 1.6 magnification changer. An AXOPATCH 200B amplifier and an Axon™ Digidata 1550A digitizer (both Molecular Devices, Sunnyvale, CA, USA) with integrated Humbug noise eliminator was used for data acquisition. Hardware was controlled with the Clampex 10.5 software (Molecular Devices). We monitored and compensated offset voltages and pipette capacitance ($C_{fast}$). Gigaohm seals were established at the cytoplasmic droplet of highly motile cells in standard high saline buffer ('HS' in mM: 135 NaCl, 20 HEPES, 10 lactic acid, five glucose, 5 KCl, 2 $CaCl_2$, 1 $MgSO_4$, one sodium pyruvate, pH 7.4 adjusted with NaOH, 320 mOsm/l) (*Kirichok et al., 2006*; *Lishko et al., 2011*). The patch pipette was filled with 140 mM $CsMeSO_3$, 20 mM HEPES, 10 mM BAPTA, 4 mM NaATP, 1 mM CsCl (pH 7.4 adjusted with CsOH, 330 mOsm/l). For recordings from capacitated spermatozoa, BAPTA was substituted for 5 mM EGTA and 1 mM EDTA. We confirmed that changing of the chelator composition had no effect on DSper current amplitudes in noncapacitated cells. Transition into whole-cell mode was achieved by applying voltage pulses (499–700 mV, 1–5 ms, $V_{hold}$ = 0 mV) and simultaneous suction. After establishment of the whole-cell configuration, inward and outward currents were elicited via 0.2 Hz stimulation with voltage ramps (−80 mV to +80 mV in 850 ms, $V_{hold}$ = 0 mV, total 1000 ms/ramp). Data was not corrected for liquid junction potential changes. To ensure stable recording conditions, only cells with baseline currents (in HS solution) ≤10 pA at −80 mV were used for experiments. Under 'HS' condition, CatSper and DSper currents were considered to be minimal, thus any remaining baseline current represented the cells leak current. During whole-cell voltage-clamp experiments, the cells were continuously perfused with varying bath solutions utilizing a gravity-driven perfusion system. If not stated otherwise,

electrophysiological experiments were performed at 22°C. Temperature of the bath solution was controlled and monitored with an automatic temperature control (TC-324B, Warner Instrument Corporation, Hamden, CT, USA). Both, CatSper and DSper currents were recorded under symmetric conditions for the major permeant ion. Under these conditions, the bath solution was divalent free ('DVF') containing (in mM) 140 $CsMeSO_3$, 20 HEPES, 1 EDTA, and pH 7.4 was adjusted with CsOH, 320 mOsm/l. To isolate DSper conductances, monovalent currents through CatSper channels were inhibited by supplementing the DVF solution with 1 mM $Mg^{2+}$ and in the absence of EDTA (*Qi et al., 2007*). Experiments with different bath solutions were performed on the same cell. Signals were sampled at 10 kHz and low-pass filtered at 1 kHz (Bessel filter; 80 dB/decade). Pipette resistance ranged from 9 to 15 MΩ, access resistance was 21–100 MΩ, membrane resistance ≥1.5 GΩ. Membrane capacitance was 0.8–1.3 pF and served as a proxy for the cell surface area and thus for normalization of current amplitudes (i.e. current density). Capacitance artifacts were graphically removed. Statistical analysis was done with Clampfit 10.3 (Molecular Devices, Sunnyvale, CA, USA), OriginPro 8.6 (OriginLab Corp., Northampton, MA, USA) and Microsoft Excel 2016 (Redmond, WA, USA). Statistical data are presented as mean ± standard error of the mean (SEM), and (n) indicates the number of recorded cells. Statistical significance was determined with unpaired t-tests.

Temperature dependency for cesium and sodium inward currents was fitted using the Boltzmann equation $y = A2 + (A1 - A2)/(1 + \exp((x - x0)/dx))$ with parameters as indicated in *Table 1*.

The temperature coefficient $Q_{10}$ reflects the temperature dependence of the membrane current and was obtained using the van't Hoff equation: $Q_{10} = (I_2/I_1)^{10/(T_2 - T_1)}$ where $I_n$ are the corresponding current amplitudes at the lower ($T_1$) and higher temperatures ($T_2$) in °C. Here, we analyzed current amplitudes at 22 and 37°C.

## Calcium imaging

All calcium imaging experiments were performed in HS solution. Prior to fluorescence recording, swim-up purified human spermatozoa were bulk loaded with 9 µM fluo-4/AM (dissolved in DMSO) and 0.05% Pluronic (dissolved in DMSO) in HS solution for 30 min at room temperature. Cells were then washed with dye-free HS solution and allowed to adhere to glass imaging chambers (World Precision Instruments, Sarasota, USA) for 1 min. Via continuous bath perfusion, the attached spermatozoa were presented with alternating extracellular conditions (HS ± agonist/antagonist; continuous presence of 1 µM NNC55-0396 as CatSper inhibitor). Fluorescence was recorded at 1 Hz, 100 ms exposure time over a total time frame as indicated. Imaging was performed using a Spectra X light engine (Lumencore, Beaverton, USA) and a Hamamatsu ORCA-ER CCD camera. Fluorescence change over time was determined as $\Delta F/F_0$ where $\Delta F$ is the change in fluorescence intensity (F - $F_0$) and $F_0$ is the baseline intensity as calculated by averaging the fluorescence signal of the first 20 s in HS solution. Regions of interest (ROI) were restricted to the flagellar principal piece of each cell by manual selection in ImageJ (Java, Redwood Shores, CA, USA). Statistical data are presented as mean ± standard error of the mean (SEM), and (n) indicates the number of recorded cells.

## Immunocytochemistry

Purified spermatozoa were plated onto 20 mm coverslips in HS and allowed to attach for 20 min. The cells were fixed with 4% paraformaldehyde (PFA) in PBS for 20 min and washed twice with PBS. Additional fixation was performed with 100% ice-cold methanol for 1 min with two washing steps in PBS. Cells were blocked and permeabilized by 1 hr incubation in PBS supplemented with 5% immunoglobulin-γ (IgG)–free BSA and 0.1% Triton X-100. Immunostaining was performed in the same blocking solution. Cells were incubated with primary antibodies (rabbit polyclonal αTRPV4, 1:100, abcam ab39260) overnight at 4°C. After extensive washing in PBS, secondary antibodies (mouse

**Table 1.** Fitting parameters.

| Permeant cation | Cell type | A1 | A2 | x0 | Dx |
|---|---|---|---|---|---|
| Cesium | noncapacitated | 0.87314 | 3.45451 | 33.8 | 6.4 |
| Cesium | capacitated | 0.90892 | 2.19096 | 31.2 | 3.7 |
| Sodium | noncapacitated | 0.84686 | 5.19918 | 34.1 | 3.6 |

DOI: https://doi.org/10.7554/eLife.35853.020

monoclonal αRabbit-DyLight488, 1:1000, Jackson 211-482-171) were added for 45 min at room temperature. After vigorous washing, cells were mounted with ProLong Gold Antifade with DAPI reagent (Life Technologies, Carlsbad, CA) and imaged with a confocal microscrope.

## RT-PCR and cloning

Total donor-specific RNA was extracted from purified spermatozoa with a QIAGEN RNAeasy mini kit followed by complementary DNA synthesis with a Phusion RT-PCR kit (Finnzymes, MA, USA). The donor-specific translated region of TRPV4 (cDNA) was amplified with the primers forward 5- ACAGA TATCACCATGGCGGATTCCAGCG −3' and reverse 5'-AACACAGCGGCCGCCTAGAGCGGGGCG TCATC-3' and was subcloned into a pTracer-CMV2 vector (Invitrogen) using the restriction sites: EcoRV and NotI. TRPV4 identity was sequence verified. HEK293 (ATCC CRL-1573) cells were transfected during passages 2 to 15 using a standard lipofectamine protocol (Invitrogen). Transfected cells were identified as green fluorescent and successful transfection was verified via both western blotting and electrophysiology. The cell lines was not tested for mycoplasma and is not on the list of commonly misidentified cell lines maintained by the International Cell Line Authentication Committee: http://iclac.org/wp-content/uploads/Cross-Contaminations-v8_0.pdf.

## Immunoblotting

The highly motile sperm fraction was separated from other somatic cells (mainly white blood cells, immature germ cells, and epithelial cells) by density gradient consisting of 90% and 50% isotonic Isolate (Irvine Scientific, CA) solution diluted in HS solution with the addition of protease inhibitors (Roche). Protease inhibitors were used throughout the whole procedure. After centrifugation at 300 g for 30 min at 24°C, the sperm pellet at the bottom of the 90% layer was collected, diluted ten times, and washed in HS by centrifugation at 2000 g for 20 min. Cells were examined by phase-contrast microscopy for motility and counted before centrifugation. Contamination of the pure sperm fraction by other cell types was minimal, with less than 0.2% of somatic cells, which was below the protein detection threshold for immunoblotting applications. The pellet was subjected to osmotic shock by a 5 min incubation in 0.5x HS solution, the addition of 10 mM EDTA and 10 mM dithiothreitol (DTT) for 10 min, and sonication in a water bath at 25°C for 5 min. Osmolarity was adjusted by addition of 10x phosphate-buffered saline (PBS). Laemmli sample buffer (5x) was added to a final 1x concentration, and the DTT concentration was adjusted to 20 mM. An additional 5 min sonication and boiling at 100°C for 5 min were performed. The total crude cell lysate was loaded onto a 4–20% gradient Tris-HCl Criterion SDS-PAGE (BioRad) with 500,000 sperm cells/well. TRPV4- and empty vector-transfected HEK293 cells were lysed in 2x Laemmli sample buffer and subjected to SDS-PAGE. Ten thousand cells per well were loaded onto SDS-PAGE. After transfer to polyvinylidene fluoride membranes, blots were blocked in 0.1% PBS-Tween20 with 3% IgG-free BSA for 15 min and incubated with primary antibodies overnight at 4°C. Blots were probed with rabbit anti-b-tubulin antibodies (Abcam), mouse monoclonal anti-actin C4 antibodies (Abcam), or anti-TRPV4 antibodies (Abcam). After subsequent washing and incubation with secondary horseradish peroxidase-conjugated antibodies (Abcam), membranes were developed with an ECL SuperSignal West Pico kit (Pierce).

## Acknowledgements

We thank Dr. Junji Suzuki (UCSF, CA) for the help with calcium imaging. We also thank Dr. Julio F Cordero-Morales (University of Tennessee Health Science Center, Memphis, TN) for the initial input and suggestions. MS is a Lichtenberg Professor of the Volkswagen Foundation and acknowledges support from the FENS-Kavli Network of Excellence. This work was supported by a DAAD fellowship to NM, and by NIH R01GM111802, Pew Biomedical Scholars Award, Alfred P Sloan Award, and Packer Wentz Endowment Will to PVL.

## Additional information

### Funding

| Funder | Grant reference number | Author |
|---|---|---|
| National Institute of General Medical Sciences | R01GM111802 | Polina V Lishko |
| Pew Charitable Trusts | 00028642 | Polina V Lishko |
| Alfred P. Sloan Foundation | FR-2015-65398 | Polina V Lishko |
| Deutscher Akademischer Austauschdienst | | Nadine Mundt |
| Packer Wentz Endowment Will | | Polina V Lishko |
| Rose Hill Innovator Fund | | Polina V Lishko |

The funders had no role in study design, data collection and interpretation, or the decision to submit the work for publication.

### Author contributions

Nadine Mundt, Conceptualization, Data curation, Formal analysis, Funding acquisition, Validation, Investigation, Visualization, Writing—original draft, Writing—review and editing; Marc Spehr, Supervision, Funding acquisition, Writing—review and editing; Polina V Lishko, Conceptualization, Resources, Data curation, Supervision, Funding acquisition, Validation, Investigation, Project administration, Writing—review and editing

### Author ORCIDs

Nadine Mundt (iD) http://orcid.org/0000-0003-3370-2933
Marc Spehr (iD) http://orcid.org/0000-0001-6616-4196
Polina V Lishko (iD) http://orcid.org/0000-0003-3140-2769

### Ethics

Human subjects: The participation of healthy human sperm donor volunteers was approved by the Committee on Human Research at the University of California, Berkeley (protocol number 2013-06-5395). All donors provided informed consent.

### Decision letter and Author response

Decision letter https://doi.org/10.7554/eLife.35853.024
Author response https://doi.org/10.7554/eLife.35853.025

## Additional files

### Supplementary files

• Transparent reporting form
DOI: https://doi.org/10.7554/eLife.35853.021

### Data availability

All data generated or analyzed during this study are included in the manuscript and supporting files. Source data files have been provided for all figures.

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
