## [Decision Letter]

Thank you for submitting your article "TRPV4 is the temperature-sensitive ion channel of human sperm" for consideration by *eLife*. Your article has been reviewed by three peer reviewers, including Leon D Islas as the Reviewing Editor and Reviewer #1, and the evaluation has been overseen by Richard Aldrich as the Senior Editor. The following individual involved in review of your submission has agreed to reveal their identity: Anne Carlson (Reviewer #2).

The reviewers have discussed the reviews with one another and the Reviewing Editor has drafted this decision to help you prepare a revised submission.

Summary:

Human sperm physiology is still not well understood. Specifically, research into the presence and molecular nature of several ion transport mechanisms is still in its infancy. It is known that sperm capacitation and hyperactivation is accompanied by an initial increase in calcium concentration mediated by CatSper channels, but the trigger of this activation, a presumed depolarization event, remained obscured. In this manuscript, Mundt at al. have undertaken a study to characterize an inward, depolarizing cationic current, that is highly sensitive to temperature and provide evidence that it is produced by the TRPV4 thermosensitive channel. The experiments are of high quality and in general support the conclusions. This paper is potentially of high general interest and impact in sperm physiology.

Essential revisions:

– The reviewers feel that it is essential that throughout the manuscript, it is made clear that the conclusions reached are so far applicable to human sperm.

– All reviewers find that the use of specific blockers, and perhaps other agonists, of TRPV4 channels is paramount to make a more solid case for TRPV4 being the basis of I_DSper_ currents.

– Since TRPV4 was cloned from the preparation used in this study, it would be greatly enhanced by showing that the properties of the cloned channels match the properties of the endogenous I_DSper_ current.

– Since TRPV1 channels are present in human sperm, although no currents with the properties of these channels have been recorded, it is essential that the authors demonstrate that no capsaicin activated currents are recorded. Please consider that these channels might be "washed-off" by the recording conditions.

Reviewer #1:

This manuscript by Mundt et al. presents evidence that the TRPV4 ion channels is responsible for a temperature-sensitive, depolarizing conductance in human sperm cells. Through careful electrophysiological and molecular biology experiments the authors describe an inward current that is separate from the main CatSper current present in these cells. The electrophysiology of sperm is riddled with controversies, and although the presence of several ion channels and transporters has been reported by inmmunoanalysis and physiology of multi-cell measurements, functional data is very hard to come by. The findings presented here are potentially of great general interest, since they might help explain the initial steps in sperm hyperactivation and offer a better understanding of sperm physiology.

The data in the manuscript is of very high quality and in general the interpretation of results is supported by the data. I have several comments and suggestions that need to be addressed.

– The authors describe a novel inward current in human sperm that they designate as I_DSper_. This is the finding in which the rest of the paper is supported and therefore it's very important that the evidence for it is strong. Although human electrophysiology is sparse, any references to this current in other reports should be included and discussed.

– If I_DSper_ is a TRPV channel, it is not clear to me why in HS solution (initial baseline recordings) the inward current is not visible, since TRPV channels are Na^+^ permeable and HS solution has a high NaCl concentration. This is a very important point to clarify.

– The authors suggest that, since I_DSper_ is temperature sensitive, it could be made up of TRPV3, TRPV4 or TRPV1. TRPV3 has a very high threshold for temperature activation, very different from I_DSper_. This should be discussed. TRPV1 is ruled out because capsaicin fails to elicit currents. However, TRPV1 protein is present in human sperm. Also, capsaicin activation is dependent of the presence of PIP_2_, since this lipid is a positive regulator of TRPV1 channels. TRPV1 cannot be activated by capsaicin at low PIP_2_ concentrations. My worry is that recording conditions might have promoted the washing off of PIP_2_ and this TRPV1 channels possibly present did not respond to capsaicin. Recordings should be made in conditions that protect the regulation of TRPV1 by PIP_2_.

– An TRPV4-selective antagonist should be used to demonstrate that I_DSper_ is indeed TRPV4.

– The authors have a nice form of evidence in that they have cloned a TRPV4 channel from human sperm. They even expressed it in HEK cells and did some western blotting. They should do electrophysiological recordings of this TRPV4 clone in HEK cells to demonstrate that is has at least some of the functional characteristics of I_DSper_.

– Temperature activation data in Figures 2 and 3 have a curve fitted to them, but no details are given as to what is its theoretical meaning. This should be fixed. Also for comparison purposes, at least a Q_10_ value should be given and discussed in terms of what is known of temperature gating in TRPV4 channels.

Reviewer #2:

Background: At mating, mammalian sperm are unable to fertilize an egg, even when the two gametes are placed in direct contact with each other. Sperm must spend time in the female reproductive tract where they acquire various physiologic processes that endow them with the capacity to fertilize, and these processes are collectively referred to as capacitation. It has long been known at the ionic currents of sperm play a commanding role in signaling capacitation, but only in the last 15 years have we begun to uncover the molecular identity of these channels.

Synopsis: Mundt and colleagues sought to uncover the identity of the channel that passes depolarizing currents during human sperm capacitation. Several pieces of data substantiated the existence of such current, including the requirement of depolarizing potentials for the proper functioning of CatSper (a channel known to be required for capacitation – hyperactivation in particular). Using electrophysiology on human sperm, the authors found that the depolarizing current: increases during capacitation, is passed by a CatSper-independent channel, increases with increasing temperatures, and can conduct Na^+^. Finally, the authors employed the TrpV4 targeting agonist, RN1747, to demonstrate that TrpV4 channels pass the depolarizing current in human sperm. These studies represent a major breakthrough in reproductive biology with interesting biological consequences.

Comments:

This paper depends on the finding that the TrpV4 agonist RN1747 is acting specifically on TrpV4 channels. Although these data are compelling, this story would be greatly strengthened by including an additionally drug targeting this channel. For example, demonstrating that TrpV4 targeting inhibitor, RN1734, blocked this current would substantiate their claim. Alternatively, is the requirement of Na-ATP (rather than Mg-ATP) TrpV4 specific? If so, perhaps that could be discussed more to substantiate these findings.

The authors focus on the physiologic temperature differences between the epididymis (34.4 °C) and body core temperature (37 °C), yet this manuscript does not demonstrate that these TrpV4 passed currents are significantly different at 34.4 vs 37 °C. Moreover, the data included in this manuscript indicate that TrpV4 channels mainly pass current after capacitation. Between the onset of hyperactivation (at the end of capacitation) and fertilization, sperm should not experience a temperature gradient. Perhaps the authors could clarify the when the gradient is experienced and when the channel is active.

Perhaps the authors could consider commenting on the role that temperature and TrpV4 may play in in vitro capacitation, which is known to require Ca^2+^, HCO_3_^-^, BSA (or another similar protein), and an elevated temperature (e.g. 36 °C). It is generally believed that capacitation cannot occur at room temperature, and a prominent role for TrpV4 may provide a mechanism for this temperature requirement.

Reviewer #3:

The manuscript by Mundt et al., describes the functional characterization of the Depolarizing Channel of Sperm (DSper). It had been previously described that hyperactivated motility necessary for the ability of sperm to fertilize an egg depends upon CatSper, the potassium cannel KSper and Hv1. Another additional Na^+^ conductance (DSper) that leads to initial membrane depolarization was also described. However, the identity of the channel responsible for this current has remained unclear. This DSper channel was investigated in the study by Mundt et al. The results here shown are solid and represent an important advance for the field of fertilization and sperm function. I have the following suggestions:

1) The Results and Discussion focus on the presence of DSper currents in human sperm. It would be interesting to discuss whether there are any observations that could account for the presence of DSper in murine models. I believe that this could help advance the field further if it was possible to perform future experiments in KO mice.

2) Figure 3D shows that DSper conducts sodium ions at both, negative and positive membrane potentials when non-capacitated sperm currents are recorded at 37°C. However, Figure 1—figure supplement 2 shows no inward sodium currents. While with magnesium, the absence of inward currents is expected, I believe that the authors should have observed sodium currents. Please explain.

3) The authors identify the nature or identity of the DSPer current being the TRPV4 channel based on, at least, two main observations: that the DSper responds to the RN1747 agonist at a concentration reported to be specific for TRPV4 activation and by cloning the channel from human sperm. In this sense, I believe it would add nicely to the manuscript if it was possible to show the block of this current by the antagonist RN1734 and by isolating the cloned channel in a heterologous expression system and performing a simple characterization of the activity of this protein.

4) Please comment on the lack of temperature-induced TRPV4 inactivation at temperatures near 35°C in the experiments shown here with respect to previous reports (Watanabe et al., 2002). Does the DSper(TRPV4) current inactivate also?

---

## [Author Response]

The major change in the manuscript is a removal of calcium imaging experiments with the TRPV4 agonist RN1747 and relying mainly on electrophysiology to confirm the activity of this agonist. The reason behind this decision is the fact that RN1747 must be dissolved in DMSO, as it is the only solvent in which RN1747 can be dissolved. DMSO (up to 0.1%) doesn’t affect DSper currents using electrophysiology, as evident from the newly added pregnenolone sulfate (PS) experiments (Figure 4—figure supplement 1E) since we have dissolved PS in DMSO as well. We found that DMSO has a profound nonspecific effect on calcium changes inside sperm cells at such small concentrations as 0.02%, as observable during calcium imaging measurements (Figure 1). This effect may not be TRPV4- related, and might not even reflect calcium influx from outside, versus release of calcium from intracellular stores. This DMSO effect has also been reported by others (Kumar et al., 2016), therefore we do not feel it is justified to rely on calcium imaging experiments when the agonist is dissolved in DMSO. All other compounds we have tested here using calcium imaging have been dissolved in ethanol.

**Author response image 1. respfig1:** Single-cell calcium imaging of the sperm flagellum. Arrow indicates application of either RN1747 dissolved in DMSO (red) or the corresponding concentration of DMSO as a vehicle control (blue). The CatSper inhibitor NNC was present during the whole recording period and did not induce a rise in cytosolic calcium levels as indicated by the black trace.

Essential revisions:– The reviewers feel that it is essential that throughout the manuscript, it is made clear that the conclusions reached are so far applicable to human sperm.

Thank you. We have made sure to clarify this and mentioned human sperm whenever applicable.

– All reviewers find that the use of specific blockers, and perhaps other agonists, of TRPV4 channels is paramount to make a more solid case for TRPV4 being the basis of I_DSper_ currents.

We agree with that, and have included additional pharmacological data, such as using two TRPV4-specific inhibitors, as well as one TRPV4-specific agonist. We have also performed comprehensive characterization of the channel cloned from sperm mRNA.

– Since TRPV4 was cloned from the preparation used in this study, it would be greatly enhanced by showing that the properties of the cloned channels match the properties of the endogenous I_DSper_ current.

Thank you. We have done this and the properties indeed match (Figure 6).

– Since TRPV1 channels are present in human sperm, although no currents with the properties of these channels have been recorded, it is essential that the authors demonstrate that no capsaicin activated currents are recorded. Please consider that these channels might be "washed-off" by the recording conditions.

Done. Thank you for this suggestion.

Reviewer #1:This manuscript by Mundt et al. presents evidence that the TRPV4 ion channels is responsible for a temperature-sensitive, depolarizing conductance in human sperm cells. Through careful electrophysiological and molecular biology experiments the authors describe an inward current that is separate from the main CatSper current present in these cells. The electrophysiology of sperm is riddled with controversies, and although the presence of several ion channels and transporters has been reported by inmmunoanalysis and physiology of multi-cell measurements, functional data is very hard to come by. The findings presented here are potentially of great general interest, since they might help explain the initial steps in sperm hyperactivation and offer a better understanding of sperm physiology.The data in the manuscript is of very high quality and in general the interpretation of results is supported by the data. I have several comments and suggestions that need to be addressed.– The authors describe a novel inward current in human sperm that they designate as I_DSper_. This is the finding in which the rest of the paper is supported and therefore it's very important that the evidence for it is strong. Although human electrophysiology is sparse, any references to this current in other reports should be included and discussed.

We have included the references to the work of others. Indeed, several temperature-sensitive ion channels or specific transporters have been reported in mammalian sperm (Gervasi et al., 2011; Hamano et al., 2016; Kumar et al., 2016). However, functional characterization of the temperature-activated cation conductance via direct methods, such as electrophysiology, has not been performed in human sperm.

– If I_DSper_ is a TRPV channel, it is not clear to me why in HS solution (initial baseline recordings) the inward current is not visible, since TRPV channels are Na^+^ permeable and HS solution has a high NaCl concentration. This is a very important point to clarify.

The residual inward current is still possible to observe in HS solution, particularly in the capacitated sperm (Figure 1B), however since TRPV4 channel is permeable to both divalent cations (calcium) and monovalent cations (sodium), there will be a competition for the pore region and the divalent block of monovalent conductance will be observed. This is a common feature shared between many calcium permeant channels, such as voltage-gated calcium channels and CatSper. For example, CatSper is perfectly permeable to calcium (Figure 1—figure supplement 1A of this manuscript), as well as sodium and cesium (Kirichok, Navarro and Clapham, 2006; Lishko et al., 2010, and Lishko et al., 2011), and yet we do not observe large CatSper currents in the solutions when both monovalent ions and calcium are present (Lishko et al., 2011 Figure 2C and Supplementary Figure 6). It has been reported that calcium exhibits a similar inhibitory influence on TRPV4 monovalent conductance with IC_50_ of 591 ± 89 nM (Watanabe et al., 2003). We have included this explanation in the manuscript.

– The authors suggest that, since I_DSper_ is temperature sensitive, it could be made up of TRPV3, TRPV4 or TRPV1. TRPV3 has a very high threshold for temperature activation, very different from I_DSper_. This should be discussed. TRPV1 is ruled out because capsaicin fails to elicit currents. However, TRPV1 protein is present in human sperm. Also, capsaicin activation is dependent of the presence of PIP_2_, since this lipid is a positive regulator of TRPV1 channels. TRPV1 cannot be activated by capsaicin at low PIP_2_ concentrations. My worry is that recording conditions might have promoted the washing off of PIP_2_ and this TRPV1 channels possibly present did not respond to capsaicin. Recordings should be made in conditions that protect the regulation of TRPV1 by PIP_2_.

We believe, the reviewer meant TRPV2, which has indeed a very high threshold for activation (above 53 °C). We have included the reference to this and the sentence describing why we have excluded TRPV2. As for TRPV1, indeed human sperm retain significant amount of TRPV1 mRNA (Miller et al., 2016), and yet we failed to detect any capsaicin-elicited activity even when we supplemented pipette solution with 30 µM PIP_2_ and applied 10 µM of capsaicin (Figure 4—figure supplement 1). It is possible that TRPV1 is initially active in the developing spermatids, and then the channel is either inactivated or removed from the plasma membrane.

– An TRPV4-selective antagonist should be used to demonstrate that I_DSper_ is indeed TRPV4.

Thank you for this suggestion. We have tested two TRPV4-selective antagonists: HC067047 and RN1734, and both prevented temperature activation of DSper, confirming that DSper is TRPV4 (Figure 5).

– The authors have a nice form of evidence in that they have cloned a TRPV4 channel from human sperm. They even expressed it in HEK cells and did some western blotting. They should do electrophysiological recordings of this TRPV4 clone in HEK cells to demonstrate that is has at least some of the functional characteristics of I_DSper_.

Thank you for this excellent suggestion. As evident from Figure 6, TRPV4 cloned from human sperm mRNA recapitulates DSper temperature sensitivity, and activation by selective agonist RN1747.

– Temperature activation data in Figures 2 and 3 have a curve fitted to them, but no details are given as to what is its theoretical meaning. This should be fixed. Also for comparison purposes, at least a Q_10_ value should be given and discussed in terms of what is known of temperature gating in TRPV4 channels.

Recombinantly expressed TRPV4 channels have reportedly high temperature coefficient Q_10_: from 9 to 19 (Guler et al., 2002; Watanabe et al., 2002). However, endogenously expressed TRPV4 recorded from aorta endothelial cells ((Watanabe et al., 2002) Figure 7A-B) have less steeper heat activation, somewhat between 1.5 and 2. In our experiments, a temperature ramp from 23°C to 37°C potentiated I_DSper_ inward currents by factors of 2.7 ± 0.5 for noncapacitated cells and 2.0 ± 0.2 for capacitated cells, which corresponds to Q_10_ noncapacitated=1.76, and Q_10 capacitated_=1.65. In our experiments we have measured Q_10_ for cesium currents (Figure 2), and not for sodium, which could account for the difference. Indeed, Q_10_ for DSper sodium conductance is higher: Q_10 noncapacitated, sodium_= 2.30 (Figure 3). Secondly, we have measured TRPV4 activation using a ramp protocol with 5 s interval between stimulations, which is a different protocol used by Watanabe et al. More importantly, as reported by Xu and colleagues, the heating speed °C/s has high impact on the resulting Q_10_ for TRPV channels (Q_10_=23.3 for 4.7 °C/s vs. Q_10_=6.4 for 2.4 °C/s for TRPV3) (Xu et al., 2002). Since we have used in-line heater to apply the change in temperature, which is a slow heating-device, it is very likely that the rather slow heating speed of our in-line heating device (0.2 °C/s) results in a low Q_10_ of DSper. It is also possible that a different lipid environment to which sperm TRPV4 is exposed, or additional modification of the channel are responsible for such differences.

Reviewer #2:Comments:This paper depends on the finding that the TrpV4 agonist RN1747 is acting specifically on TrpV4 channels. Although these data are compelling, this story would be greatly strengthened by including an additionally drug targeting this channel. For example, demonstrating that TrpV4 targeting inhibitor, RN1734, blocked this current would substantiate their claim. Alternatively, is the requirement of Na-ATP (rather than Mg-ATP) TrpV4 specific? If so, perhaps that could be discussed more to substantiate these findings.

Thank you for this suggestion. We have tested two TRPV4-selective antagonists: HC067047 and RN1734, and both prevented temperature activation of DSper, confirming that DSper is TRPV4 (Figure 5). As for sodium vs magnesium ATP, indeed, there is a requirement for this as according to Phelps et al. (Phelps, et al., 2010) since intracellular Na-ATP binding to the N-terminal ankyrin repeat domain of TRPV4 has a profound sensitizing effect, which is a feature of the TRPV ankyrin repeats and is shared between TRPV1 and TRPV4 (Lishko et al., 2007). Indeed, addition of 4 mM ATP to the pipette solution, allowed us to consistently record TRPV4 activity from fertile human sperm.

*The authors focus on the physiologic temperature differences between the epididymis (34.4* °*C) and body core temperature (37* °*C), yet this manuscript does not demonstrate that these TrpV4 passed currents are significantly different at 34.4 vs 37* °*C. Moreover, the data included in this manuscript indicate that TrpV4 channels mainly pass current after capacitation. Between the onset of hyperactivation (at the end of capacitation) and fertilization, sperm should not experience a temperature gradient. Perhaps the authors could clarify the when the gradient is experienced and when the channel is active.*

This is an excellent suggestion, and we have included the paragraph discussing this in the Discussion.

*Perhaps the authors could consider commenting on the role that temperature and TrpV4 may play in in vitro capacitation, which is known to require Ca^2+^, HCO_3_^-^, BSA (or another similar protein), and an elevated temperature (e.g. 36* °*C). It is generally believed that capacitation cannot occur at room temperature, and a prominent role for TrpV4 may provide a mechanism for this temperature requirement.*

This is an excellent suggestion, and we have included the paragraph discussing this in the Discussion.

Reviewer #3:The manuscript by Mundt et al., describes the functional characterization of the Depolarizing Channel of Sperm (DSper). It had been previously described that hyperactivated motility necessary for the ability of sperm to fertilize an egg depends upon CatSper, the potassium cannel KSper and Hv1. Another additional Na^+^ conductance (DSper) that leads to initial membrane depolarization was also described. However, the identity of the channel responsible for this current has remained unclear. This DSper channel was investigated in the study by Mundt et al. The results here shown are solid and represent an important advance for the field of fertilization and sperm function. I have the following suggestions:1) The Results and Discussion focus on the presence of DSper currents in human sperm. It would be interesting to discuss whether there are any observations that could account for the presence of DSper in murine models. I believe that this could help advance the field further if it was possible to perform future experiments in KO mice.

Thank you for this suggestion. We have included the following paragraph: “It is also possible that sperm cells possess more than one type of temperature-activated TRP-like ion channels. The biphasic inhibition of DSper with TRPV4-selective antagonists (Figure 5) does not result in complete current inhibition, particularly in the temperate range between 24°C and 32°C. This may suggest an additional, non-TRPV4 conductance. The molecular nature of such additional conductance(s) could be a temperature-sensitivity of CatSper to its inhibitors NNC 55-0396, or perhaps the presents of other temperature-sensitive ion channel(s). Interestingly, according to one published report (Hamano et al., 2016), murine TRPV4 regulates sperm thermotaxis. However, TRPV4-deficient male mice are fertile which may indicate either presence of an additional temperature sensor or a compensatory mechanism.

*2) Figure 3D shows that DSper conducts sodium ions at both, negative and positive membrane potentials when non-capacitated sperm currents are recorded at 37*°*C. However, Figure 1—figure supplement 2 shows no inward sodium currents. While with magnesium, the absence of inward currents is expected, I believe that the authors should have observed sodium currents. Please explain.*

It is possible that cesium and sodium currents have a different permeability through TRPV4, especially since Cs is a larger ion. Sodium inward currents will be therefore larger than Cs inward currents – consistent with what we observe.

3) The authors identify the nature or identity of the DSPer current being the TRPV4 channel based on, at least, two main observations: that the DSper responds to the RN1747 agonist at a concentration reported to be specific for TRPV4 activation and by cloning the channel from human sperm. In this sense, I believe it would add nicely to the manuscript if it was possible to show the block of this current by the antagonist RN1734 and by isolating the cloned channel in a heterologous expression system and performing a simple characterization of the activity of this protein.

Thank you for this suggestion. We have tested two TRPV4-selective antagonists: HC067047 and RN1734, and both prevented temperature activation of DSper, confirming that DSper is TRPV4 (Figure 5). We have also confirmed the physiology of the cloned channel (Figure 6).

4) Please comment on the lack of temperature-induced TRPV4 inactivation at temperatures near 35°C in the experiments shown here with respect to previous reports (Watanabe et al., 2002). Does the DSper(TRPV4) current inactivate also?

Since we have measured TRPV4 activation using a ramp protocol with 5s interval between stimulations, which is a different protocol used by Watanabe et al., it is possible that the initial fast activation/inactivation phase of DSper was not reflected in our recording, and we have focused on its steady-state phase. However, according to Watanabe et al., endogenous TRPV4 kinetic, recorded from aorta endothelial cells (Watanabe et al., 2002 JBC) does not have fast activation/inactivation kinetics, which might be a result of different lipid environment to which sperm TRPV4 is exposed, or additional modification of the endogenous channel.